# Performance Analysis of Magnetorheological Porous Fabric Composite

**Hua Yan** [1,*], **Lifan Wu** [2,*], **Pingyang Li** [2], **Xuan Li** [2], **Junxin Zhao** [2] and **Xiaomin Dong** [2]

1    School of Intelligent Manufacturing Engineering, Chongqing College of Architecture and Technology, Chongqing 401311, China
2    State Key Laboratory of Mechanical Transmission, Chongqing University, Chongqing 400044, China
*    Correspondence: yanhua@cqrec.edu.cn (H.Y.); wulifan@stu.cqu.edu.cn (L.W.)

**Abstract:** In this paper, magnetorheological (MR) porous fabric composites were prepared by using non-woven fabrics pretreated with lithium stearate. Due to the porous structure of non-woven fabric and the good stability of lithium stearate, the mechanical properties and sedimentation stability of the composite have been improved. The viscosity, shear stress and dynamic viscoelasticity of different samples were analyzed by rheological properties tests. The results indicate that the mechanical properties of the composite samples containing lithium stearate are improved compared with the MRF (magnetorheological fluid) sample, and its sedimentation stability under non-working conditions is also improved. However, with the increase of lithium stearate content, the improving effect of material performance gradually decreased. The experimental results show that when the magnetic flux density is 0.31 T, the shear stress of the MR porous fabric composite with 1 wt% lithium stearate is about 27 kPa, which increases by 51.1% compared with the MRF sample.

**Keywords:** magnetorheological fluid; rheological properties; composite; pretreatment

## 1. Introduction

Magnetorheological (MR) composite, which is composed of magnetorheological fluid (MRF) and other materials such as fabric and sponges, is a new group of smart material [1–4]. Due to a reversible change of certain mechanical and rheological properties under the application of an external magnetic field, MR composites exhibit superior performances and are applied in many engineering applications, such as polishing [5], aeronautic [6] and other industrial fields [7,8]. In previous literature, the development and preparation of MR composites mainly include MRF composites [9–12], MR elastomers [13–17] and MR fabric composites [18–21]. For example, in order to offer guidelines for the preparation of soft composites with tunable field-dependent Young's moduli, Yoon et al. [22] investigated mechanical performances of soft composites made from MR elastomer and MRF considering the effect of Young's modulus. To explore the mechanical theory of magneto-sensitive materials, Ivaneyko et al. [23] analyzed a regular rectangular lattice model of magneto-sensitive elastomers in a homogeneous magnetic field. It is found that the shear modulus of all particles increases with the increase of the magnetic field. Karl et al. [24] analyzed MR elastomers with switchable mechanical properties and the influence of different components on them. The experimental results show that the modification of the particle surface by silane can significantly improve the hardness and tensile strength of the materials. Moreover, the composite filled with carbonyl iron has a better controllable elastic modulus. Kaleta et al. [25] presented MR elastomers with isotropic and anisotropic structure and their manufacturing methods, where magneto-mechanical properties of the MR composites are examined. It provides the basis for the fatigue research of MR composites in the future. Małecki et al. [26] prepared an MR composite consisting of block copolymer styrene-ethylene-butylene-styrene and carbonyl iron powder and described its dynamic mechanical behavior. The results show that the hysteresis loops' area

of the modified carbonyl iron powder composites is smaller than that of the conventional composite. Son et al. [27] manufactured an MRF-augmented fabric to enhance the ballistic performance of fragment barrier materials. The experimental results demonstrated that the composite has a good ability to absorb impact energy.

Based on the above-mentioned, there are numerous studies on preparation, development and application of MR composites; only a few articles reported the aspect of using MRF with porous structures. The porous structure has a better effect on the mechanical properties of composites [28]. Presently, the most widely used fabrics are woven from aramid and ultrahigh molecular weight polyethylene (UHMWPE) fibers. For example, Mistik et al. [29] prepared and tested three-dimensional spacer fabric with MRF to improve the stiffness of the material. Moreover, the stiffness of the composite excited by the magnetic field is significantly improved at ambient temperature. With the increase in temperature, the stiffness decreases, but the thermal conductivity increases. Glaser et al. [30] presented the preparation and test of three types of composites, including MR fabric composites, magnetic elastomers and MR sponge composites, in order to determine the possible rigidification properties of the magnetic and MR materials. The results illustrate that the magnetic particles or MR components in the composite samples have a great influence on the performance of the device, and the properties are related to the content of magnetic particles and the surface state. Non-woven fabric is a new kind of MR carrier substrate, which has the capability of preventing leakage of the MRF in the rotary devices and effectively enhances mechanic performances and sedimentation stability of MRF [31]. Nakano et al. [32] developed a small-scale seismic linear MR damper by using the MRF porous composite made of non-woven porous materials impregnated with MRF to prevent sedimentation of magnetic particles. The results show that the damping force of the designed damper can reach 20 kN at the applied current of 0.5 A. Dong et al. [33] proposed a vibration absorber based on a variable inertia flywheel by employing MRF stored in non-woven fabric, where experimental results are in good agreement with the simulation. Moreover, the inertia adjustment range of the proposed inertia flywheel can reach 27.2%.

Although non-woven fabrics with MRF have been widely researched and explored, most literature focuses on applications based on non-woven fabrics and lacks analysis of some factors affecting their mechanical properties. Thus, the main work of the paper is to investigate the effect of pretreatment on output mechanical performance of non-woven fabric.

In this paper, in order to improve the performance of MRF, non-woven fabric is selected as the key material, and lithium stearate is used to pretreat it so as to prepare MR porous fabric composite. The mechanical properties of the composite are evaluated experimentally by testing the rheological properties of different samples. The viscosity, dynamic yield stress and dynamic viscoelasticity of different samples are analyzed in detail by comparison. Finally, by comparing the mechanical properties of the pretreated samples and the MRF sample, the effect of the pretreatment on the material properties is evaluated, and the feasibility of pretreatment to improve the performances of MR porous fabric composite is verified.

## 2. Preparation and Experiments

In this section, MR porous fabric composite has been prepared with different treatments. Meanwhile, comparative experiments of MR porous fabric composite have been carried out to analyze the mechanical properties.

### 2.1. Material Preparation

In the preparation of MR porous fabric composite, the commercial MRF (MRF-G28; Chongqing Materials Research Institute Co., Ltd., Chongqing, China) is used as the working medium. The maximum shear stress of this MRF is larger than 50 kPa. The non-woven fabric (Chang Tai Non-woven Fabric Co., Ltd., Shenzhen, China) has been selected as the matrix to store MRF. Material parameters of MRF and non-woven fabric have been listed in Tables 1 and 2. Figure 1 shows a photo of the non-woven fabric used.

**Table 1.** Parameters of MRF-G28.

| Property | Values | Property | Values |
|---|---|---|---|
| Density/kg·m$^{-3}$ | 2960 | Shear stress at 0.5 T/kPa | $\geq$50 |
| Viscosity of zero-field ($\dot{\gamma}$ = 51/s, 20 °C)/Pa·s | $\leq$1.5 | Operating temperature/°C | $-40 \sim 150$ |
| Bulk modulus/MPa | 200 | Magnetic properties/kA·m$^{-1}$ | 379.64 |

**Table 2.** Parameters of non-woven fabric.

| Ingredient | Density/kg·m$^{-3}$ | Average Thickness/mm |
|---|---|---|
| Polypropylene | 200 | 0.01 |

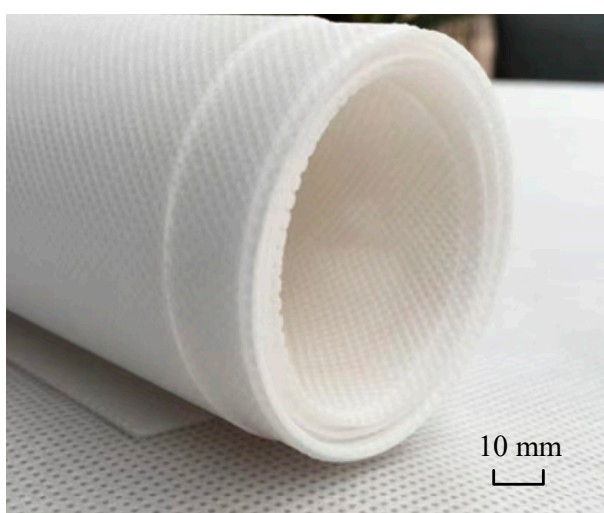

**Figure 1.** Photograph of non-woven fabric.

The MR porous fabric composite is prepared through a three-step process. Firstly, the non-woven fabric has been cleaned. The commercial non-woven fabric is cut into small-size pieces, which is the same as the testing area of the rheometer. Then, non-woven fabrics with a fixed size are immersed in 95% ethanol solution (Shanghai Aladdin Co., Ltd., Shanghai, China) at room temperature for 2 h. The immersed fabric is dried at 80 °C for 1 h. This step is used to remove the impurities. In the second step, the cleaned non-woven fabrics will be pretreated. All cleaned samples are divided into three parts equally. These cleaned samples are immersed into lithium stearate (LiC$_{18}$H$_{35}$O$_2$, as a commonly used stabilizer in industry, has excellent thermal stability; Purity: 90%; Shanghai Aladdin Co., Ltd., Shanghai, China) by ultrasonic dispersion to ensure the required homogeneity and promote combinations between lithium stearate and internal fibers. Figure 2 shows the non-woven fabric being pretreated. The three beakers were respectively filled with 1 wt%, 1.5 wt% and 2 wt% lithium stearate solution for soaking non-woven fabric. Thirdly, the pretreated non-woven fabrics are mixed into the special MRF for 1 h by ultrasonic dispersion. The extra MRF of the treated non-woven fabric is separated by centrifugation. Moreover, three test samples containing different mass fractions of lithium stearate were prepared. Therefore, the pretreated MR porous fabric composite can be prepared with different pretreatments.

The pretreated MR porous fabric composite can form the internal porous network to store MRF and protect particle chains. It can reinforce the performance compared with other conventional MRF.

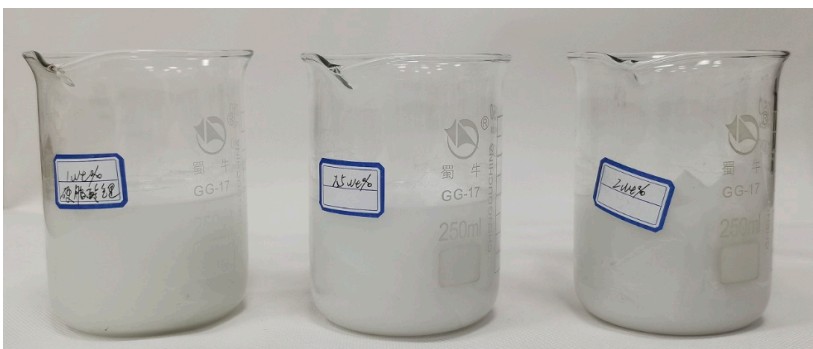

**Figure 2.** Non-woven fabric being pretreated (The left is the soaking solution with lithium stearate content of 1 wt%, the middle is 1.5 wt% of that, and the right is 2 wt% of that).

## 2.2. Experiments

Mechanical properties of the MR porous fabric composite have been tested by comparative experiments. The experimental setups are shown in Figure 3. The commercial rheometer (Physica MCR 302, Anton Paar Co. Ltd., Graz, Austria) has been used to measure the rheological properties of the tested samples. The use of parallel plate rotation tests groove and PP20 type plate rotor (diameter is 20 mm). During the test, the sensor probe was 1 mm away from the liquid level of the material (under zero-field conditions). At the same time, in order to improve the accuracy of the test, the water bath circuit was used to control the temperature of the test device, and the temperature of the test environment was controlled at a stable value. Figure 3a shows the experimental system of MR porous fabric composite. Through the control platform to set the test instructions, the magnetic field control unit will be given a specific magnetic flux density. Then the rheometer starts to test the rheological properties of the material and upload the test results to the control platform.

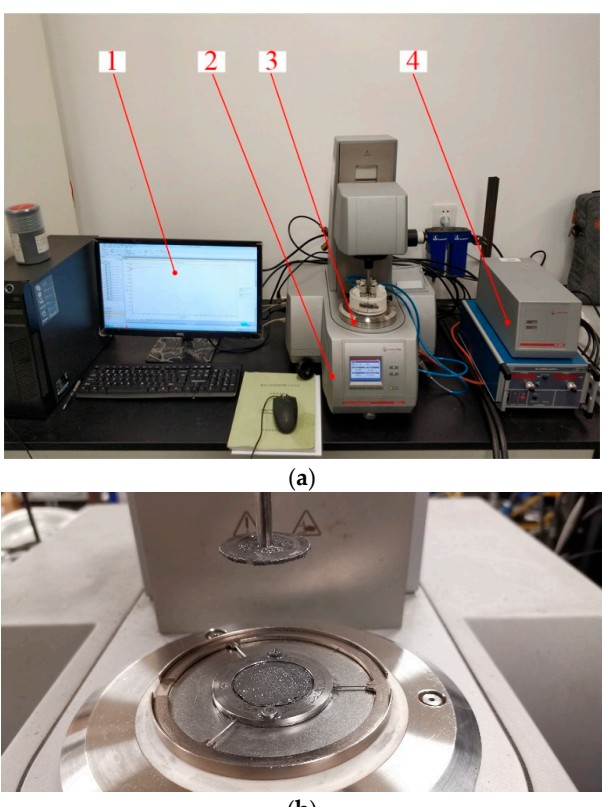

(**a**)

(**b**)

**Figure 3.** Experimental set-ups of the MR porous fabric composite. (**a**) Experimental system: (1) control platform, (2) rheometer, (3) test platform, (4) magnetic field control unit. (**b**) main body of test.

In order to deeply analyze the performance of the proposed MR material, the steady shear and dynamic oscillatory modes of the rheometer have been carried out. Figure 3b shows the test body of the test. In the steady shear mode, the pretreated samples are placed in the measuring region between the parallel plate and electromagnet. The test temperature is fixed at 25 °C by the thermostatic system. The shear rate changes from 0.1 to 100 s$^{-1}$. The shear stress and viscosity at different shear rates were obtained by applying magnetic flux density from 0 to 0.75 T to the sample.

In the dynamic oscillatory mode, the shear strain sweep mode is used to measure the dynamic properties of MR porous fabric composite. The applied magnetic flux density changes from 0 T to 0.6 T with an increment of 0.3 T. The storage modulus and loss modulus can be measured at various magnetic flux densities.

## 3. Discussion

### 3.1. Viscosity Changeability

Figure 4 shows the curve of viscosity variation with the shear rate of the test sample under zero-field conditions. As an important index to evaluate the performance of MRF, viscosity affects its output characteristics. It can be seen from Figure 4 that the apparent viscosity of different test samples decreases with the increase of shear rate, showing the phenomenon of shear thinning. The MRF at zero-field and zero shear rate behaves as a liquid state because it is not excited by the magnetic field, and its apparent viscosity is small, about 8.3 kPa·s. As the control group, the composite of untreated non-woven fabric and MRF showed an apparent enhanced viscosity compared with MRF under the zero-field and zero shear rate, and the initial viscosity of the 2 wt% sample group was about 34.2 kPa·s.

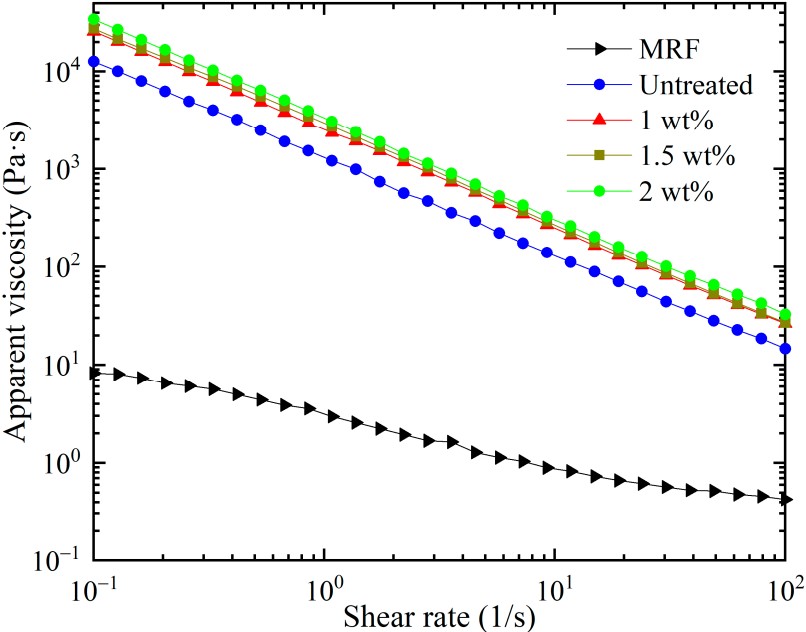

**Figure 4.** Variation of viscosity with shear rate under zero-field.

The results show that the addition of non-woven fabric in the composite has a thickening effect, and the flow of suspended particles is hindered to a certain extent due to the three-dimensional mesh structure of the non-woven fabric and the lithium stearate attached to it. The peak viscosity of each test sample in the non-shearing state also indicates that each sample has high viscosity characteristics in the non-working state, which can ensure the sedimentation stability of the fluid material. With the increase in shear rate, particle clusters in the MRF composite samples were largely decomposed under the action of a high shear rate, resulting in a significant decrease in the viscosity of each sample. However, the downward trend is roughly the same, which also indicates that the addition of non-woven

fabric has no great influence on the controllable characteristics of MRF, which provides a certain basis for the subsequent application of this composite.

As can be seen in Figure 4, the viscosity of the composite samples after pretreatment is generally larger than that without pretreatment. With the increase of a mass fraction of lithium stearate, the viscosity of the composite is also improved. Compared with the MRF sample, the viscosity of the other three groups of samples increased significantly due to the addition of non-woven fabric. The main reason is that the magnetic particle clusters without the MR effect are dispersed significantly and cannot form a chain-like structure. In this case, the viscosity is dominated by the non-woven fabric and lithium stearate.

Figure 5 shows the viscosity curves of the five test samples under different magnetic flux densities. The increase of lithium stearate content cannot improve the viscosity of the composite with the increase of magnetic flux density. Moreover, the viscosity improvement rate of the composite even shows a downward trend. It can be seen from Figure 5 that when the magnetic flux density increases from 0.31 T to 0.6 T, the viscosity improvement rate of the composite obviously decreases. Further, the viscosity of some samples is even lower than that of MRF at 0.6 T. The main reason for this phenomenon was that the non-woven fabric was soaked in lithium stearate solution, and a certain amount of lithium stearate particles were adsorbed on the surface, which increased the internal viscosity of the composite, thus impeding the internal flow of the fluid and improving its viscosity. However, when the mass fraction of lithium stearate is too large, the mass fraction of magnetic particles in the composite will be relatively reduced. Moreover, the viscosity of the composite is dominated by the chain-like structure formed by the magnetic particles under the action of a strong magnetic field, which obstructs the movement of the fluid and thus exhibits certain viscosity characteristics.

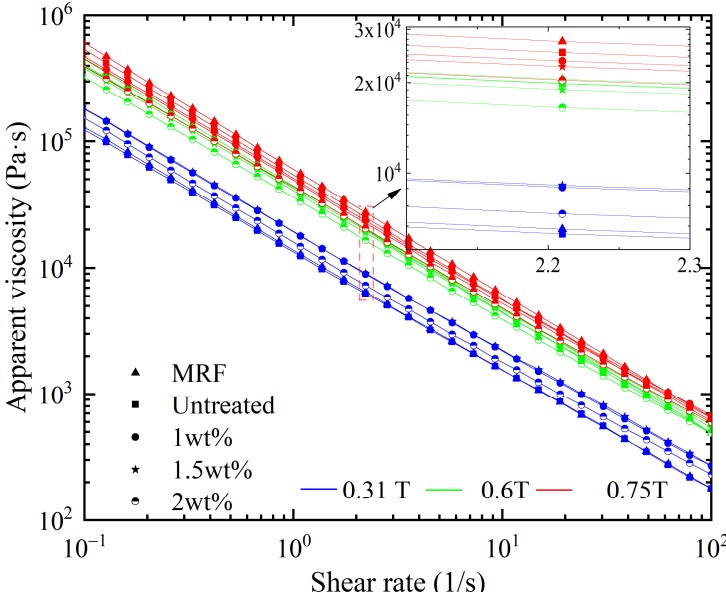

**Figure 5.** Variation of viscosity with shear rate under different magnetic flux densities.

According to the experimental results, when the magnetic flux density is 0.31 T, the initial viscosity values of the MRF, untreated (0 wt%), 1 wt%, 1.5 wt% and 2 wt% samples (Xwt% represents the sample group of composite with lithium stearate content of Xwt%, the same as below) are 131.7, 125.2, 183.4, 185.5 and 156.3 kPa·s, respectively. Therefore, appropriate immersion treatment of non-woven fabrics with lithium stearate can improve the viscosity of composite under medium and low magnetic fields.

### 3.2. Shear Stress

Figure 6 shows the variation curves of shear stress with the shear rate of each test sample under zero-field conditions. Because there is no rheological effect in MRF, its shear

stress is mainly provided by the shear friction of the material itself. The maximum shear stress is about 42.8 Pa, which almost does not change with the shear rate, and the liquid shows the characteristics of Newton fluid.

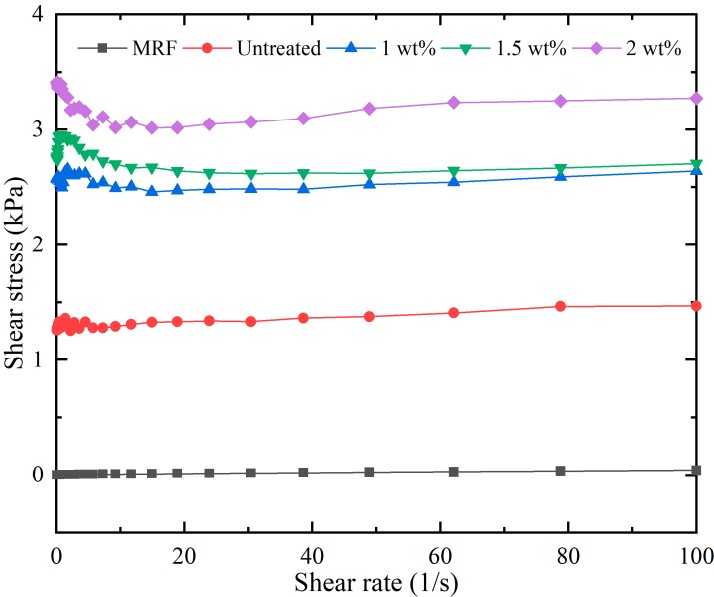

**Figure 6.** Variation of shear stress with shear rate under zero-field.

The untreated sample group (the lithium stearate content is 0 wt%) also changed similarly to the MRF group, but due to the porous structure of the non-woven fabrics, the mechanical properties of the untreated sample were improved to a certain extent, and the maximum shear stress is about 1.5 kPa. For the samples treated by immersion, the shear stress increases greatly under zero-field and increases with the increase of the mass fraction of lithium stearate. This is mainly because the samples are not affected by the magnetic field, and the shear stress is dominated by the friction between particles scattered inside the material. With the increase of the mass fraction of lithium stearate, the shear stress also increases.

In order to further analyze the effect of the pretreatment for non-woven fabric on the composites, the output properties of each sample under different magnetic fields were tested. Figure 7 shows the variation curves of shear stress with the shear rate of four composite samples under different excitations, and the variation laws are roughly the same.

At a low shear rate, the rheological effect occurs under the excitation of the magnetic field, and the shear stress of each sample increases greatly. The rheological effect makes the magnetic particles form a chain-like structure along the direction of the magnetic field, and the structure becomes more stable with the increase of the magnetic flux density. When the sample material is subjected to shear action, the chain-like structure formed by magnetic particles will undergo a dynamic "break-recover" process, thus showing the effect of shear stress.

For the untreated sample, the maximum shear stress is about 61.1 kPa. As can be seen from Figure 7, when the non-woven fabric is soaked in lithium stearate solution, the mechanical properties of the composites do not always improve with the increase of the mass fraction of lithium stearate. Consistent with the apparent viscosity of the material, the shear stress of the 1 wt% sample is higher than that of the untreated sample, and the maximum shear stress is about 67.6 kPa. Under the same working condition, the shear yield strength of the 1 wt% sample is higher than that of the untreated sample, and the highest yield strength is increased by 52%. As the mass fraction of lithium stearate continued to increase, the mechanical properties of the samples began to decrease. The shear stress of the 1.5 wt% and 2 wt% samples are 64.7 kPa and 59.4 kPa, respectively.

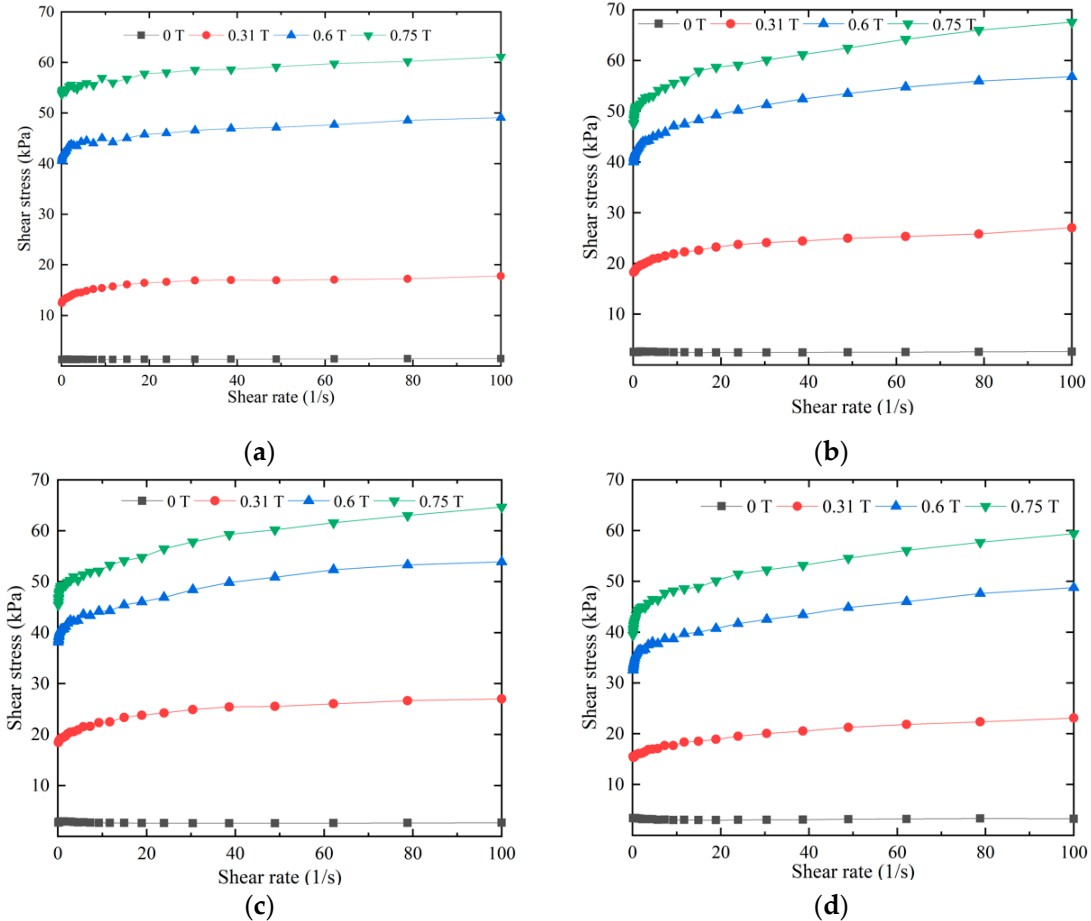

**Figure 7.** Variation of shear stress with shear rate of four composite samples under different excitations. (**a**) the 0 wt% sample, (**b**) the 1 wt% sample, (**c**) the 1.5 wt% sample and (**d**) the 2 wt% sample.

It is obvious from the experimental results that the mechanical properties of the samples will not continue to improve with the increase of lithium stearate mass fraction of the non-woven fabric. When the mass fraction of lithium stearate increases to 2 wt%, the shear stress decreases compared with the 1 wt% sample. Mainly because with the increase of lithium stearate content, a part of the lithium stearate attached to the surface of non-woven fabric will be filled in the micro-porous structure of non-woven fabric, thus reducing the effect of non-woven fabric.

Figure 8 shows the shear stress comparison curve of the MRF, untreated and 1 wt% samples. As can be seen from the figure, the shear stress of the untreated sample is enhanced compared with that of the MRF sample at zero-field due to the material properties of the non-woven fabric. However, when the magnetic flux density increases, its mechanical properties are not different from those of the MRF sample or even inferior to those of the MRF sample.

However, the mechanical properties of 1 wt% sample pretreated with non-woven fabrics are significantly improved. When the magnetic flux density is 0.31 T, the maximum shear stress is about 27 kPa, which increases by 51.1% compared with the MRF sample under the same working conditions. With the increase of magnetic flux density, the enhancement effect of mechanical properties is slightly weakened. When the magnetic flux density is increased to about 0.75 T, the shear stress of the MRF sample is even greater than the 1 wt% sample at lower shear rates. Therefore, from the perspective of mechanical properties, it can be concluded that appropriate pretreatment for non-woven fabrics can improve the mechanical properties of composites.

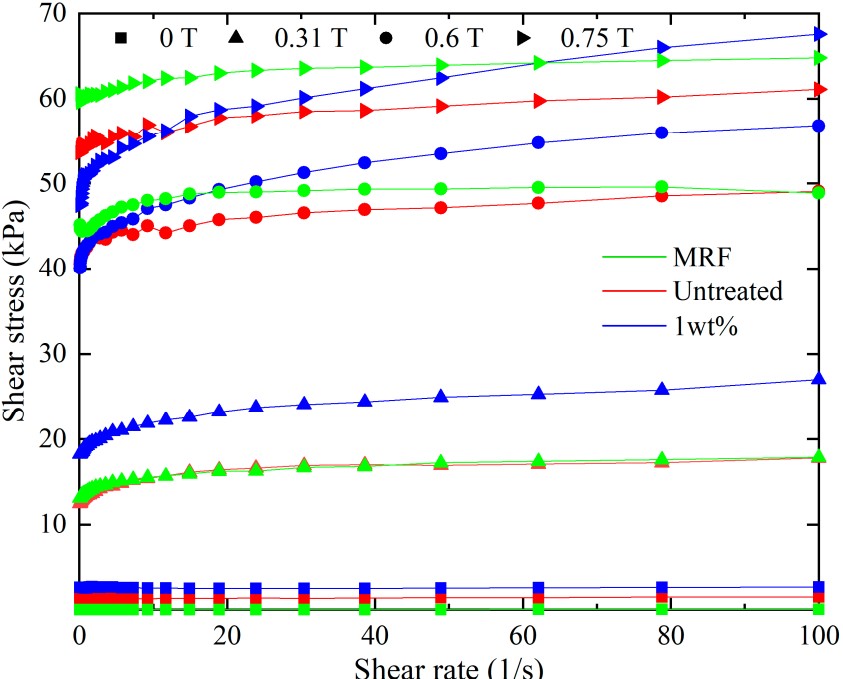

**Figure 8.** Comparison of shear stress between different samples.

It can be seen from the experimental data that the mechanical properties of the 1 wt% sample are better than those of the other three groups of control samples (the 0 wt%, 1.5 wt%, 2 wt% samples), and their mechanical properties are greatly improved compared with the MRF sample. Therefore, lithium stearate with appropriate mass fraction can optimize the mechanical properties of the composite under low and medium magnetic flux density.

*3.3. Dynamic Viscoelasticity*

The storage modulus $G'$ and loss modulus $G''$ are the key parameters to characterize the dynamic viscoelasticity of MRF materials. The storage modulus is mainly affected by the force between magnetic particles generated by the MR effect, which belongs to elastic energy. The loss modulus is affected by the energy dissipation of friction between different phases when the material is sheared [34]. In general, MRF can be approximated as a linear viscoelastic material under small shear strain. Its dynamic shear moduli can be equivalent to the complex modulus $G^*$ related to the excitation magnetic field $H$ and the angular frequency $\omega$. The shear stress $\tau$ of the material can be expressed as follows:

$$\tau(t) = G^*(H, \omega)\gamma(t) \tag{1}$$

where the complex modulus $G^*$ is:

$$G* = G\prime + iG'' \tag{2}$$

Therefore, the friction of fluid and the motion of magnetic particles can be neglected under the low shear strain and fixed oscillation frequency, and the energy dissipation of MRF material is mainly the friction between magnetic particles. According to Equations (1) and (2), the shear stress of materials can be regarded as magnetic stress $\tau_H$ and friction stress $\tau_f$, which can be expressed as follows:

$$\tau = \tau_H + \tau_f \tag{3}$$

Among them:

$$\tau_H(t) = G\prime(H, \omega)\gamma(t) \tag{4}$$

$$\tau_f(t) = G''(H, \omega)\gamma(t) \tag{5}$$

In order to further test the sample with 1 wt% lithium stearate, dynamic viscoelastic experiments were carried out. Figure 9 shows the variation curve of dynamic shear moduli with shear strain at a frequency of 10 rad·s$^{-1}$ for the 1 wt% sample. As can be seen from the figure, the dynamic shear moduli of the sample decrease with the increase of strain amplitude under the zero field, and the storage modulus decays faster. When the shear strain exceeds a critical value (about 2.2%), the storage modulus is less than the loss modulus, so the composite is in a flow state.

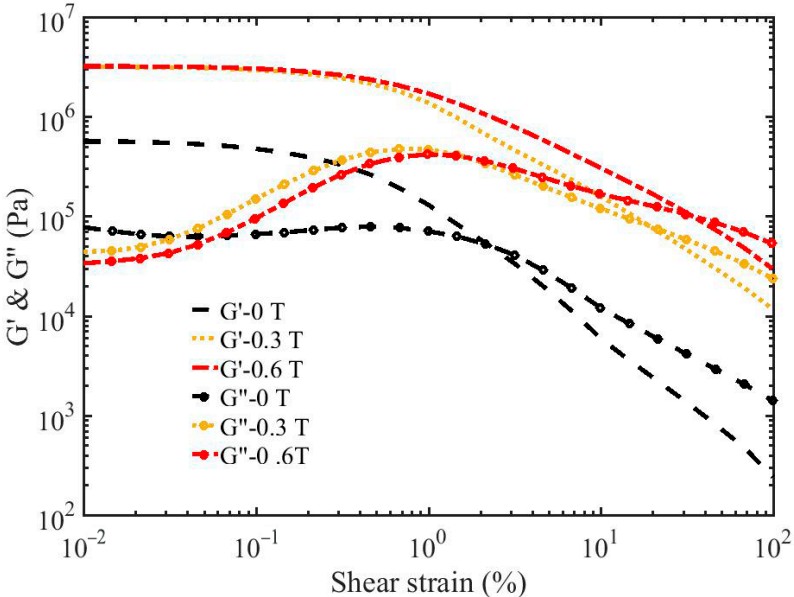

**Figure 9.** Dynamic shear moduli at the constant frequency.

With the continuous increase of the excitation magnetic field, the storage modulus of the composite has been significantly improved, and it is maintained at a relatively stable value under a low strain amplitude. When the shear strain continues to increase to a certain threshold, the storage modulus begins to decline. As for the loss modulus, it has an upward trend and then a downward trend when the shear strain is low. At this time, the composite presents viscoelastic. This threshold divides the MRF material into linear viscoelastic (LVE) and nonlinear viscoelastic (NLVE). It can be seen from the figure that the dynamic shear moduli of the sample increase with the increase of the magnetic flux density, and the range of the linear viscoelastic also increases slightly. It is mainly due to the rheological effect inside the composite that makes the magnetic particles form a more stable chain-like structure with the increase of the magnetic flux density, resulting in viscoelasticity.

The chain-like structure of the magnetic particles starts to be destroyed when the shear strain exceeds the threshold. At this time, this structure is in the dynamic equilibrium stage of "break-recover". When the shear strain continues to increase, the dynamic equilibrium of the structure is broken, and the composite is in a flowing state. It can also be seen from Figure 9 that the storage modulus of the 1 wt% sample is relatively large under a magnetic field. The higher storage modulus of 1 wt% samples is mainly attributed to its higher viscosity, which makes the chain-like structure formed by the rheological effect more stable.

## 4. Conclusions

In this paper, the properties of MR porous fabric composite are studied. The non-woven fabric was pretreated with lithium stearate and mixed with commercial MRF by ultrasonic dispersion. Finally, different MR porous fabric composite samples were prepared. The influence of pretreatment on the properties of composites was studied by rheological tests.

Under the zero-field condition, the maximum shear stress of the MRF sample is 42.8 Pa, while the 2 wt% sample is up to 3.3 kPa. After pretreatment, the viscosity and shear stress of the samples are greatly increased. When the magnetic flux density is 0.31 T, the shear stress of the MR porous fabric composite with 1 wt% lithium stearate is about 27 kPa. Compared with the MRF sample, the shear stress of the 1 wt% sample is increased by up to 51.1%.

The experimental results show that the appropriate content of lithium stearate can improve the rheological properties of the composite and its sedimentation stability in the non-working state. Specifically, the movement and sedimentation of magnetic particles in the composite are hindered by the microscopic porous structure of the non-woven fabric and the good stability of lithium stearate. However, with the increase of lithium stearate content, the mechanical properties of the composite are not improved significantly.

The chain formation mechanism of MR materials is an important research content. Due to the limitation of experimental conditions, we have not conducted in-depth research on the micromorphology of composite. Therefore, it is necessary to carry out theoretical and experimental research on the chain formation mechanism of materials in the future.

**Author Contributions:** H.Y. contributed the preparation of the MR composites and edited the paper; P.L. and L.W. conducted the experimental research; L.W. wrote the first draft; X.D. carried out the theoretical analysis and revised the paper; X.L. and J.Z. supported the experimental process. All authors have read and agreed to the published version of the manuscript.

**Funding:** This research received no external funding.

**Data Availability Statement:** Not applicable.

**Conflicts of Interest:** The authors declare no conflict of interest.

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
