# Peer review of "Performance Analysis of Magnetorheological Porous Fabric Composite"

_applsci, doi:10.3390/app122312062_

Round 1

Reviewer 1 Report

Reviewer comments:

Manuscript number: applsci-2030846

(Performance Analysis of Magnetorheological Porous Fabric Composite)

The paper illustrates the  magnetorheological porous fabric composites prepared by using non-woven fabrics pretreated with lithium stearate

The paper is interesting, however before considering it suitable for publication in the journal some suggestions should be incorporated in the manuscript.

Remarks:

1.     What is main novelty in present work?

2.     English grammatical mistakes and some typo mistakes should be removed.

3.     What is total cost of experiments?

4.     When the content of lithium stearate is 1wt%, the performance of the composite is greatly improved. What is possible reason for this behavior?

5.     Compared with the MRF sample, the shear stress of the composite is increased by up to 51.1% 312 at the magnetic flux density of 0.31 T. What is possible reason for this behavior?

6.     Conclusion should be written pointwise for better readability.

7.     Some recent work should be included in manuscript.

Flexural analysis of laminated composite porous plate. Asian Journal of Civil Engineering 2022 (https://doi.org/10.1007/s42107-022-00523-y)

Reviewer 2 Report

Authors are required to address the following comments:

·    Introduction:

1.     Instead of writing “applied on many engineering problems”, specify the application and refer to each application of MR fluids individually in the sentence.

2.     In the introduction part, authors need to address the results or numerical data reported in the literature by various authors.

·        Discussion

1.       Dynamic viscoelasticity should also be reported and compared for 1.5wt% and 2w% lithium stearate also.

2.       Page no. 5, line no 171; Check this sentence. Provide a value of magnetic field instead of writing “certain value”.

3.       In Fig. 4; why the viscosity reduction rate of MRF is not linear (others are linear)?

4.       Authors should develop the numerical modeling using curve fitting for the obtained viscosity plots.

Reviewer 3 Report

Dear authors,

Thank you for your manuscript. This is a very interesting issue that you are working on. However, there are still some changes that need to be taken into account:

-Apart from references 13-16, the authors should mention more publication MR elastomers related articles (applications and modeling) since they play an important application on various fields. Please summarize references 13-16 and the following literature in a few sentences:

Carlson, J.D., Jolly, M.R.; MR fluid, foam and elastomer devices; (2000) Mechatronics, 10 (4), 555-569.

Karl, C.W.; McIntyre, J.; Alshuth, T.; Klüppel, M.; Magneto-rheological elastomers with switchable mechanical properties; (2013) KGK Kautschuk Gummi Kunststoffe, 66 (1-2), 46 - 53

Ivaneyko, D., Toshchevikov, V.P., Saphiannikova, M., Heinrich, G.; Magneto-sensitive elastomers in a homogeneous magnetic field: A rectangular lattice model, (2011) Macromolecular Theory and Simulations, 20 (6), 411-424. - Please provide more information in the introduction about the different applications and technical limitations for the materials you have investigated. 

- Section 2.1 Material preparation: Please provide more details about the composition of the used material (commercial MRF) and about the non- woven fabric. How has the non-woven fabric been pretreated (see line 91)?

- Fig. 1: Please provide a scale bar in this picture.

- All the used chemicals should be mentioned in a separate section (composition, purity and supplier). Why did you select lithium stearate?

- Please give a detailed description of Fig. 2: What are the differences of the shown beakers? Please provide more details about the pretreatment procedure.

- Fig. 3 b: How is the experiment carried out? Give more details in the text. Mention Fig. 3 a and b in the text accordingly.

- Please show repeated experiments regarding the results in Fig. 4. Why did you only choose 2w.-%? Please describe it clearly for the reader. 

- Please explain to the reader why the damping effect of non-woven fabric can be almost ignored (see lines 161/162).

- The inset in Fig. 5 is very difficult to read. Please improve the resolution.

- The conclusion part is too short. You should provide a discussion about the results, the consequences and the limitations. It would be good to have a short outlook part what you would recommend experiments for further investigations.

Round 2

Reviewer 2 Report

The authors have responded to all the comments raised earlier. The manuscript can be accepted for publication. 

Reviewer 3 Report

Dear authors,

Thank you for the revised form and your changes. I have no more annotations.